# High Purity, Crystallinity and Electromechanical Sensitivity of Lead-Free (Ba_0.85_Ca_0.15_)(Zr_0.10_Ti_0.90_)O_3_ Synthesized Using an EDTA/glycerol Modified Pechini Method

**DOI:** 10.3390/ma18051180

**Published:** 2025-03-06

**Authors:** Laura Caggiu, Costantino Cau, Marzia Mureddu, Stefano Enzo, Fabrizio Murgia, Lorena Pardo, Sonia Lopez-Esteban, Jose F. Bartolomé, Gabriele Mulas, Roberto Orrù, Sebastiano Garroni

**Affiliations:** 1Department of Chemical, Physical, Mathematical and Natural Sciences, University of Sassari, Via Vienna 2, 07100 Sassari, Italy; lcaggiu@uniss.it (L.C.); c.cau1@phd.uniss.it (C.C.); m.mureddu6@studenti.uniss.it (M.M.); enzo@uniss.it (S.E.); fmurgia@uniss.it (F.M.); mulas@uniss.it (G.M.); 2Department of Architecture, Design and Urban Planning, University of Sassari, Piazza Duomo 6, 07041 Alghero, Italy; 3Instituto de Ciencia de Materiales de Madrid (ICMM), Consejo Superior de Investigaciones Científicas (CSIC), c/Sor Juana Inés de la Cruz, 3, Cantoblanco, 28049 Madrid, Spain; lpardo@icmm.csic.es (L.P.); s.lopez.esteban@csic.es (S.L.-E.); jbartolo@icmm.csic.es (J.F.B.); 4Department of Mechanical, Chemical and Materials Engineering, University of Cagliari, Via Marengo 2, 09123 Cagliari, Italy; roberto.orru@unica.it

**Keywords:** Ba_0.85_Ca_0.15_Ti_0.9_Zr_0.1_O_3_, lead-free piezoceramics, sol–gel synthesis, microstructure, EDTA-glycerol

## Abstract

A single (Ba_0.85_Ca_0.15_)(Zr_0.10_Ti_0.90_)O_3_ phase material with a tetragonal structure is processed and synthesized with a modified Pechini method using ethylenediaminetetraacetic acid and glycerol as chelating and esterifying agents, respectively. The complete chemical transformation to the desired phase is achieved at 900 °C, which is 300 °C lower than conventional synthesis methods. Its consolidation, reaching up to 91% relative density, is carried out at 1400 °C. It is clearly demonstrated that the use of ethylenediaminetetraacetic acid and glycerol reagents is particularly beneficial for inducing a homogeneous grain size distribution (10 μm), which leads to very promising electromechanical properties (d_33_ = 451 pC/N; d_31_ = 160 pC/N; kp = 0.40; $\varepsilon \prime_{33}^{T}$ = 4790 and Q_m_ = 358) of the densified system.

## 1. Introduction

Piezoceramics are pivotal components for many key technologies, such as actuators, ultrasonic motors, sensors, etc. [1]. Among lead-free candidates, barium calcium zirconate titanate (BCZT), which is based on non-hazardous elements, has become one of the most extensively studied systems due to its high d_33_ values (620 pC/N) [2]. The piezoelectric properties of BCZT arise from the presence of coexisting phases at the “polymorphic phase boundary” (PPB), a transition region in the composition–temperature phase diagram that separates two different crystallographic phases, which confer excellent electromechanical properties. For this system, the optimal composition has been found to be BZT-50 BCT, namely (Ba_0.85_Ca_0.15_)(Zr_0.10_Ti_0.90_)O_3_ [3,4]. On the other hand, the BCZT preparation shows an important drawback represented by the high processing temperatures: 1300 °C and 1450–1550 °C for the synthesis and sintering steps, respectively [2,5,6]. The synthetic route has a great influence on the final microstructure that leads to the enhancement of the electromechanical properties. Therefore, sustainable and efficient synthesis that allows for modulating the microstructure and crystallite size distribution is mandatory for this class of materials [7]. In the sol–gel technique, the ceramic precursors are dispersed in solution, permitting optimum mixing that promotes a more homogeneous composition, better control of stoichiometry and crystallite size and, as a direct consequence, a drastic decrease in the synthesis temperatures [7,8,9,10,11]. The Pechini method is a widely recognized approach within sol–gel synthesis techniques, particularly valued for its versatility in obtaining homogeneous multicomponent oxide materials. In the modified Pechini method, citric acid (CA) is introduced as a chelating agent, playing a crucial role in preventing metal ion segregation in the resulting product. Citric acid mediates the interactions between metal ions and the solution by partially replacing alkoxy groups with carboxylate ligands. This substitution facilitates a stable and uniform coordination environment around the metal ions, thus promoting homogeneity and reducing phase separation during synthesis [12]. However, the effectiveness of citric acid as a chelating agent is highly dependent on the solution’s pH. To address this limitation, a modified Pechini method has been developed for the synthesis of BCZT, in which citric acid is substituted with ethylenediaminetetraacetic acid (EDTA). This modification provides enhanced stability across a broader pH range, facilitating improved control over the synthesis process and product properties [13]. Concerning the esterifying agents, glycerol offers several advantages over the commonly used, but toxic, ethylene glycol: it combines the advantages of water (low toxicity, low cost, and high availability) and ionic liquids (high boiling point and low vapor pressure) [14,15,16,17]. Recently, Aziguli et al. [18] investigated the correlation between the improvement of the electromechanical properties and the addition of glycerol in the synthesis of the BZT-50 BCT system. It was reported that the resulting BZT-50 BCT pellet, prepared from the powders synthesized by using glycerol, exhibits more homogenous crystallite size distribution and good piezoelectric and dielectric performance.

In this work, an alternative and more efficient modified Pechini method combining glycerol and EDTA is proposed for the synthesis of the BZT-50 BCT system. The combination of EDTA and glycerol led to a narrow grain size distribution and good electromechanical properties.

## 2. Materials and Methods

**Sample preparation.** The BCZT precursors, used in a stoichiometric ratio to obtain the (Ba_0.85_Ca_0.15_)(Zr_0.10_Ti_0.90_)O_3_ phase, were barium carbonate (CAS 513-77-9; 99%), calcium acetate (CAS 5743-26-0; 99%), zirconium (IV) butoxide (CAS 1071-76-7) and titanium tetraisopropoxide, or TTIP (CAS 546-68-9; 97%). Ethylenediaminetetraacetic acid (EDTA) (CAS 60-00-4; 99.4%), glycerol (CAS 56-81-5; 99.5%) and ammonium hydroxide (CAS 7664-41-7) were used as chelating, esterifying and pH-corrective agents, respectively. To compare the effect of combined glycerol/EDTA (principal route: BCZT_EDTA_), a second synthesis (named BCZT_CA_) was conducted, with the substitution of EDTA with citric acid (CA) (CAS 77-72-9; 99.5%). The principal synthesis has been performed following a parallel reaction scheme (BC and ZT in Figure 1): specifically, once 30 mL of water was heated at 60 °C, in the principal synthesis, 13.151 g of EDTA with 13 mL of ammonium hydroxide was added to two separate beakers. After 30 min, 0.839 g of barium carbonate with 0.199 g of calcium acetate (BC) and 0.23 mL of zirconium butoxide with 1.35 mL of TTIP (ZT) were introduced in the corresponding EDTA/H_2_O solutions, as schematized in Figure 1.

The BC and ZT solutions were merged in a controlled and gradual manner; 1 h later, the ammonium hydroxide was added. Subsequently, the system was annealed at 80 °C for 1 h and glycerol was added into the solution in a second step to facilitate the gelification. The final gel was then calcined in the muffle (NEYO series 2) at 900 °C for 2 h with a heating ramp of 5 °C/min. After the calcination, the material appears as a white, light and fragile agglomerate.

The as-synthesized powders were mixed with PVA solution (3 wt.%) and pressed into disk-shape pellets using a uniaxial press (220 Kg/cm^2^ for 30 min) on a diameter die. The green bodies were sintered in air using a horizontal oven (Nabertherm) with an intermediate step performed to eliminate the binder at 600 °C for 2 h, followed by a final sintering plateau at 1400 °C for 2 h, with a 3 °C/min heating rate. Bulk densities were measured by employing a geometrical method. The sintered ceramics were reduced in thickness by polishing down to 1 mm. Silver paste was deposited on both disk surfaces and then sintered (400 °C/1 h) to obtain the Ag electrodes. The samples were poled in thickness under an electric field of 15 kV/cm for 30 min at 50 °C in a silicone oil bath, followed by field cooling to room temperature.

**Characterization.** After the polarization, the quasi-static measurement of the piezoelectric constant (d_33_) was carried out using a Berlincourt d_33_-meter at 100 Hz (Sinocera YE2730). The measurement was repeated with an interval of 2 h to highlight some possible differences.

The standardized resonance method [19] was used to determine the electromechanical, elastic and dielectric coefficients of the ceramics, together with the electromechanical coupling factor. Symbols of all material coefficients determined here correspond to the standard definitions, namely: piezoelectric charge coefficients (d_ij_, where ij = 31 and 33), planar electromechanical coupling factor (k_p_), planar frequency number (N_p_), loss Q factor (Q_i_, where i = p states for the piezoelectric coefficient and i = m states for the mechanical coefficients), the real part of the free dielectric permittivity (*ε*^′^_33_^*T*^) and dielectric losses (tan δ). Standard also defines the sample shape and aspect ratios to be used for this measurement [19]. According to the standard, complex impedance, expressed in terms of modulus and phase ((/Z/,θ) plot), as a function of frequency, the so-called resonance curve, was measured with an impedance analyzer (HP 4192A-LF, Hewlett-Packard, Palo Alto, CA, USA) at the electrically induced radial extensional resonance of the thickness poled thin disks.

An automatic iterative method, advantageous with respect to the standard calculations [19], was used in the analysis of the resonance curves to determine the material coefficients in complex form, i.e., including all losses, whose full details can be found elsewhere [20]. The standard calculation is based on the frequencies of the minimum and maximum impedance modulus as well as the interval between these two frequencies to determine the real part of the material coefficients, the dielectric loss tangent and one mechanical loss factor (Q). For the sake of this automatic iterative calculation, an alternative representation of the complex impedance in terms of the peaks of resistance and conductance ((R,G) peaks plot) is calculated from the measured data. From the peaks of the resistance and conductance, four characteristic frequencies are automatically determined. The non-linear analytical expression of the complex impedance at the resonance as a function of the frequency, of the material parameters, of the sample dimensions and of the sample density is considered for the iterative calculation. Using the measured complex impedance values at these four characteristic frequencies, a system of four non-linear equations is established at each step of the iterative process, and this system is numerically solved until a convergence criterion is reached to obtain the complex material parameters. Afterwards, the (R,G) curves are recalculated using the analytical expression and the calculated material coefficients. A regression factor, ℜ^2^, of the recalculated experimental curves is obtained as a quality criterion of the material parameters. The closer ℜ^2^ is to 1, the more reliable the parameters.

For the sake of this calculation, an alternative representation of the complex impedance in terms of the peaks of resistance and conductance ((R,G) plot) was used. Permittivity vs. temperature curves at frequencies above 1 kHz (1, 2, 5, 10, 20, 50, 100, 500 and 100 kHz) were performed using automatic temperature control and capacitance-loss tangent data acquisition from an impedance analyzer (HP 4194 A).

The X-ray powder diffraction (XRD) measurements were conducted at room temperature with a θ/2θ Bragg–Brentano diffractometer (SmartLab Rigaku, Tokyo, Japan.) equipped with a copper rotating anode (CuKα_1_ = 1.5406 Å, CuKα_2_ = 1.5443 Å). Data were acquired step-wise in a 2θ range from 10° to 120°, with a step size of 0.05° and with a fixed dwell time of 4 s. The XRD patterns were numerically refined by the Rietveld technique using MAUD software [21], the COD (Crystallographic Open Database) and Pearson’s Crystal Data for the structure Crystallographic Information Format (CIF) files [22]. The fresh fracture surface morphology of the samples was studied using scanning electron microscopy (FEI Quanta 200). Backscattered electrons (BSEs) were employed to highlight and compare the processed materials. SEM instrument was equipped with EDS Microanalysis (Genesis XM2i-Apollo 10SSD LNfree).

## 3. Results and Discussion

The X-ray diffraction patterns of the BCZT_EDTA_ powders are shown in Figure 2 together with the control synthesis performed with citric acid (BCZT_CA_). The results obtained from the Rietveld analysis of the two syntheses are reported in Table 1.

The product synthesized starting from EDTA shows a BCZT tetragonal phase (# 1532166, S.G.: *P*4*mm*) as the major phase (98 wt. %) characterized by a c/a of 1.003 and a 2 wt.% of impurity attributed to CaTiO_3_ (# 1000022, S.G.: *Pbnm*). The pattern collected on the BCZT_CA_ powders presents a higher percentage of impurity, resulting 3% of unreacted ZrO_2_ (# 1533766, S.G.: *P*42*/nmc*) and 97 wt.% of the BCZT perovskite phase characterized by a slightly higher tetragonal distortion of 1.006 (c/a with c = 4.024 Å), consistent with the literature data [23].

From a microstructural point of view, both systems show an anisotropic distribution of the coherent domain size for the BCZT phase (Table 1), with crystallite dimensions of 980 nm and 820 nm for the BCZT_EDTA_ and BCZT_CA_ systems, respectively.

The introduction of EDTA in the synthesis process significantly reduces the calcination temperature by 100 °C, enabling the formation of a highly pure BCZT tetragonal phase (>98 wt.%) at 900 °C [24,25].

The powders were subsequently sintered at 1400 °C for 2 h and polarized. The pellet was achieved with a uniaxial press using a 1.27 cm-diameter die, while the final diameters, accounting for shrinkage during sintering, measured 1.10(48) cm for BCZT_EDTA_ and 1.11(73) cm for BCZT_CA_. Each pellet was then polished to a thickness of 1 mm. The geometrically determined density of the final disks (Figure 3a) reached 91% of the theoretical density (5.78 g/cm^3^) for BCZT_EDTA_ and 80% for BCZT_CA._

The diffraction patterns of the BCZT_EDTA_ ceramic (BCZT_EDTAc_) and BCZT_CA_ ceramic (BCZT_CAc_), reported in Figure 3b, exhibit a single tetragonal phase (*P*4*mm*). In the BCZT_EDTAc_ pattern, the high-angle peaks were more sharply resolved (see zoom-in view in Figure 3c), indicating a higher tetragonal distortion with respect to BCZT_CAc_. In fact, BCZT_EDTAc_ presents a c/a ratio of 1.003 (c = 4.015 Å), while BCZT_CAc_ shows a c/a ratio of 1.001 (c = 4.022 Å), as schematically represented in Figure 3d and reported in Table 2.

This significant difference in the c/a ratio can be attributed to the further reaction that occurred during the sintering process between the unreacted ZrO_2_ and the BCZT tetragonal phase. Computational and experimental studies, in fact, evidenced that the diffusion of Zr^4+^ (ionic radius 0.72 Å) in the Ti^4+^ (ionic radius: 0.605 Å) sites within the BCZT lattice is favored at high temperatures, leading to a reduction in the c/a ratio as the a lattice parameter increases [26,27].

**Figure 3 materials-18-01180-f003:**
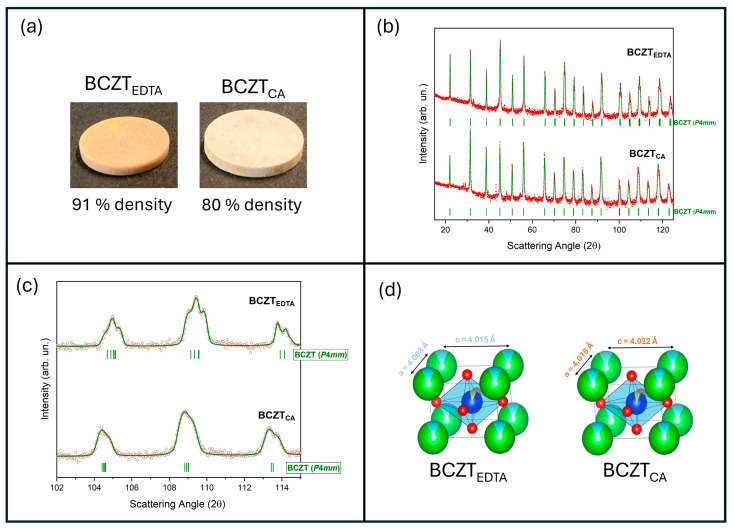
(**a**) Sintered ceramics with the corresponding densities. (**b**) X-ray diffraction patterns of the ceramics produced from the powders synthesized with EDTA and CA. Green full lines indicate the Bragg reflections and profiles of the *P*4*mm* BCZT phase. The experimental pattern is represented by red dots. (**c**) Enlarged view of the peaks in the 102°–115° 2θ angular range. (**d**) 3D vista (obtained by VESTA software [28]) of the elementary BCZT cell with a and c lattice parameters. Green, light blue, red, blue and yellow refer to barium, calcium, oxygen, titanium and zirconium atoms, respectively.

The micrograph collected on the fracture surface of the BCZT_CAc_ system (Figure 4a,b) showed some pores, probably caused by an incomplete densification process that corroborated the non-optimal sample density (80%). The pore size distribution was not unimodal, with pores ranging from 3.8 to 11.6 µm. The grain sizes varied from 2.5 µm to 10 µm, evidencing a non-homogeneous distribution. On the contrary, BCZT_EDTAc_ (Figure 4c,d) showed a transgranular fracture, indicating proper densification. The pore size distribution was nearly unimodal, with an average pore size of 3.5 (± 0.5) µm.

To further support the structural and microstructural analysis, Energy Dispersive X-ray (EDX) mapping was performed to evaluate the elemental homogeneity of the samples. The EDX analysis confirmed a more uniform distribution of Ba, Ca, Zr, and Ti in the BCZT_EDTAc_ sample compared to BCZT_CAc_. Figure 5 shows the elemental mapping for both samples, highlighting a homogeneous dispersion of Zr and Ti in the BCZT_EDTAc_ system, whereas the BCZT_CAc_ sample exhibited localized inhomogeneities. This result aligns well with the XRD and SEM observations, reinforcing the superior densification and phase purity of BCZT_EDTAc_

As expected from the structural and morphological characterizations, the poled BCZT_CAc_ pellets did not show piezoelectric activity. Conversely, the measurement at the d_33_ meter and the measured complex impedance curves, reported in Figure 6 as R and G plots, on the poled BCZT_EDTAc_ ceramic, evidenced the high value of piezoelectric coefficients (d_33_ = 451 pC/N and d_31_ = −160 pC/N) and high electromechanical coupling factor (k_p_ = 0.40) (Table 3).

Figure 7 shows the dielectric permittivity as a function of the temperature at increasing frequency. The clear dielectric anomaly, which presents the maximum of the permittivity in the same range as previously reported results, confirms the ferroelectric–paraelectric phase transition [29].

Due to the homogeneous microstructure achieved, the mechanical losses obtained are low as the mechanical quality factor is high (Q_m_ = 343). These properties are comparable to those obtained in similar synthesis studies. Preveen et al. achieved a higher d_33_ value (530 pC/N), although this required elevated calcination (110 °C) and sintering (1550 °C) temperatures [30]. Ji et al. measured a similar d_33_ value (488 pC/N) using an autoclave and the carcinogenic reagent ethoxyethanol [29].

A deeper analysis of the electromechanical response suggests that the observed d_33_ value of 451 pC/N in BCZT_EDTAc_ is competitive with the best-performing lead-free piezoceramics reported in the literature [31]. Unlike the systems requiring higher processing temperatures, the modified Pechini method employed here allows for the formation of highly dense and well-textured ceramics at significantly lower thermal budgets. This improvement not only enhances energy efficiency in material processing but also reduces undesirable exaggerated and inhomogeneous grain growth, which is often linked to microstructural and performance degradation in piezoceramics. Moreover, the enhanced electromechanical coupling factor (k_p_ = 40.35%) is indicative of an optimized domain structure, likely facilitated by the homogeneity in grain size and elemental distribution confirmed via EDX.A strong correlation between homogenous and dense microstructure, and improved poling efficiency has been observed in recent studies, reinforcing the importance of a controlled synthesis approach [32]. Compared to traditional sol-gel and solid-state methods, the Pechini-derived BCZT_EDTAc_ system exhibits enhanced stability across a broader pH range, facilitating improved control over the synthesis process, which leads superior balance between structural integrity and functional performance. Additionally, the high dielectric permittivity ($\varepsilon \prime_{33}^{T}$= 4790) coupled with a low loss tangent (tan δ = 0.007) ensures minimal energy dissipation, a crucial factor for applications in energy harvesting and actuators. The stability of these properties over time and temperature cycles will be the focus of future investigations, as understanding their long-term reliability is critical for commercial adoption.

## 4. Conclusions

In conclusion, this study demonstrates, for the first time to our knowledge, the feasibility of the processing of the lead-free Ba_0.85_Ca_0.15_Ti_0.9_Zr_0.1_O_3_ with promising electromechanical properties, synthesized using a modified Pechini method where citric acid (pH sensible) and ethylene glycol (toxic) were replaced by EDTA and glycerol, respectively. The XRD analysis comparison of the two sintered compounds indicates that BCZT synthesized using EDTA–glycerol combination displays a higher degree of tetragonal distortion than BCZT synthesized with citric acid, which enhances the piezoelectric properties of BCZT_EDTA_. Furthermore, the sintered pellet processed, which is characterized by a single tetragonal phase (*P*4*mm*), reveals an optimal microstructure with a homogeneous grain size distribution centered around 10 μm and excellent electromechanical coefficients (d_33_ = 451 pC/N; d_31_ = −160 pC/N; k_p_= 0.40; $\varepsilon \prime_{33}^{T}$ = 4790 and Q_m_ = 343). The adopted approach is more valuable as compared to that proposed so far, where toxic reagents and higher temperatures for calcination are required. This result opens new perspectives for the sustainable synthesis and processing of highly performing BCZT lead-free piezoceramics.

## Figures and Tables

**Figure 1 materials-18-01180-f001:**
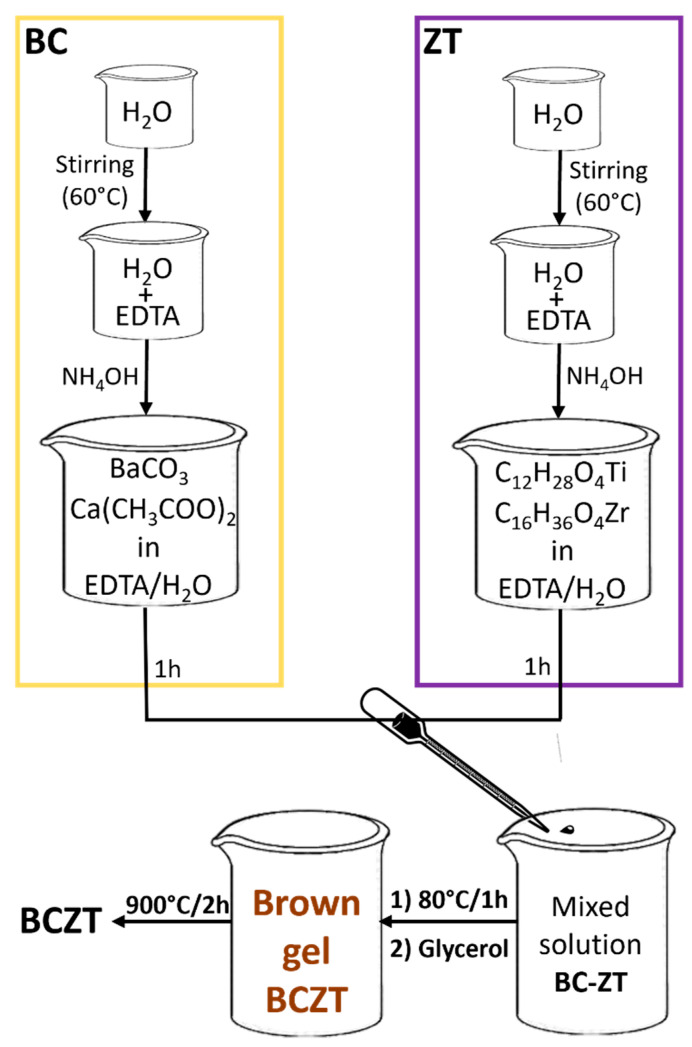
Scheme of the synthesis route. In the BC panel, barium carbonate and calcium acetate have been used as metal precursors. In the ZT panel, zirconium butoxide and TTIP have been used as metal precursors. A second synthesis attempt was performed replacing EDTA with citric acid.

**Figure 2 materials-18-01180-f002:**
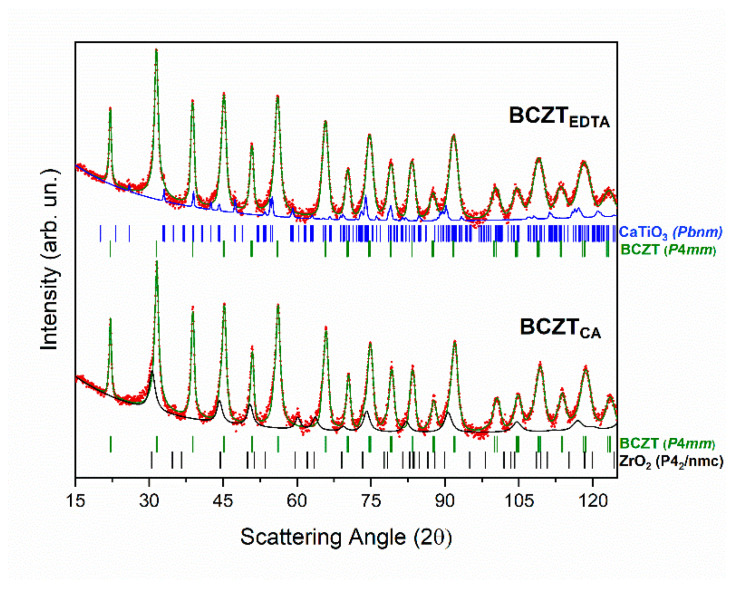
Experimental XRD patterns of the BCZT_EDTA_ and BCZT_CA_ calcined powders. Experimental points are indicated with red dots. BCZT, ZrO_2_ and CaTiO_3_ phases are reported using green, black, and blue full lines, respectively. RWp%: 7.6 (pattern BCZT_EDTA_). RWp%: 6.5 (pattern BCZT_CA_).

**Figure 4 materials-18-01180-f004:**
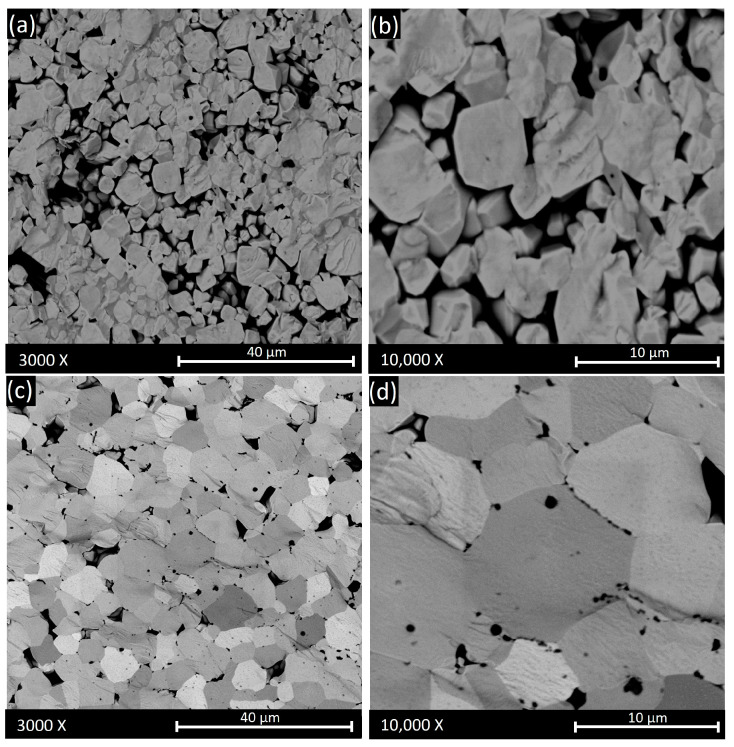
SEM images acquired using a backscattered electron detector (BSDE) on the fracture surfaces of the sintered BCZT_CAc_ (**a**,**b**) and BCZT_EDTAc_ (**c**,**d**) pellets.

**Figure 5 materials-18-01180-f005:**
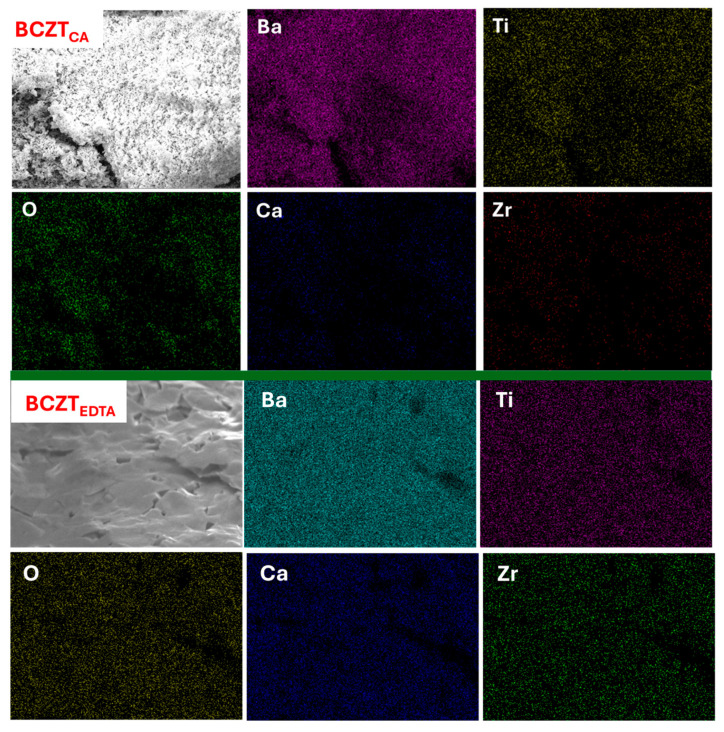
EDX elemental mapping of BCZT_CAc_ and BCZT_EDTAc_ ceramics.

**Figure 6 materials-18-01180-f006:**
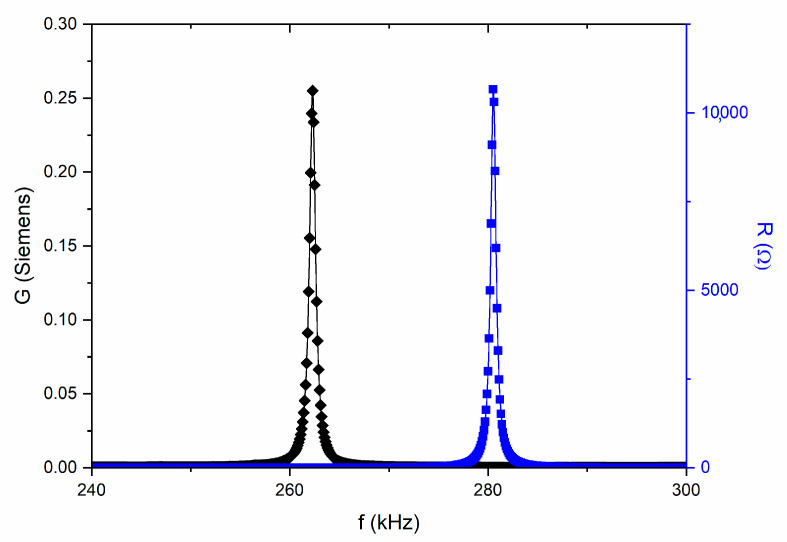
**Figure 6.** (R, G) plot, used in the calculation of material coefficients using the iterative automatic method [20]. Symbols are the experimental data and lines are the reconstructed peaks after coefficients calculation. Fundamental radial mode of resonance of a thin disk of BCZT_EDTAc_ ceramics.

**Figure 7 materials-18-01180-f007:**
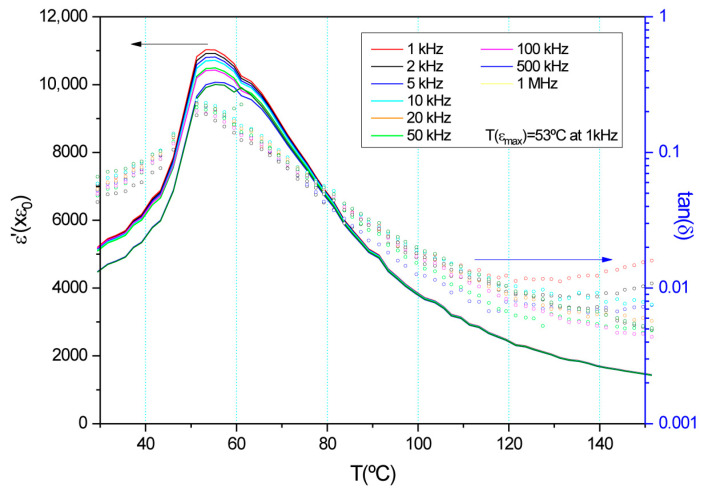
Dielectric permittivity vs. temperature for BCZT_EDTAc_ ceramic.

**Table 1 materials-18-01180-t001:** Lattice and microstructural parameters calculated using the Rietveld method from the XRD experimental pattern of the calcinated materials.

BCZT_EDTA_	Composition (wt.%)	a (Å)	b (Å)	c (Å)	Cryst.Size (Å)	rms
**BCZT** (S.G.: *P*4*mm*)	98	4.015	4.015	4.027	980	4.0 × 10^−4^
**CaTiO_3_** (S.G.: *Pbnm*)	2	5.400	5.457	7.654	990	8.9 × 10^−4^
**BCZT_CA_**	**Composition (wt.%)**	**a (Å)**	**b (Å)**	**c (Å)**	**Cryst.Size (Å)**	**rms**
**BCZT** (S.G.: *P*4*mm*)	77	4.008	4.008	4.024	820	3.7 × 10^−4^
**ZrO_2_** (S.G.: *P*4_2_/*nmc*)	23	3.621	3.621	5.001	440	2.0 × 10^−3^

**Table 2 materials-18-01180-t002:** Lattice and microstructural parameters calculated using the Rietveld method from the XRD experimental pattern of the sintered powders.

BCZT_EDTA_	Composition (wt.%)	a (Å)	b (Å)	c (Å)	Cryst.Size (Å)	rms
**BCZT** (S.G.: *P*4*mm*)	100	4.003	4.003	4.015	>2000	6.9 × 10^−4^
**BCZT_CA_**	**Composition (wt.%)**	**a (Å)**	**b (Å)**	**c (Å)**	**Cryst.Size (Å)**	**rms**
**BCZT** (S.G.: *P*4*mm*)	100	4.017	4.017	4.022	>2000	9.4 × 10^−4^

**Table 3 materials-18-01180-t003:** Relevant material coefficients obtained by iterative analysis of the impedance curve at planar resonance and d_33_ charge coefficient measured using a Berlincourt meter for the BCZT_EDTAc_ system.

Properties	
**Density (g/cm^3^)**	5.26
**d_33_** **(pC/N)**	451 (440 after 2 h)
ℜ^2^	0.9986
**k_p_ (%)**	40.35
**N_p_(** **kHz·mm** **)**	2898
**d′_31_ (pC/N)**	−160
**Q_p_(d_31_)**	113
$c\prime_{11}^{p}$ **(10^10^N m^−2^)**	9.8
**Q_m_**	343
$\varepsilon \prime_{33}^{T}$	4790
**tan δ**	0.007

## Data Availability

The original contributions presented in this study are included in the article. Further inquiries can be directed to the corresponding author.

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
