# Peer review of "High Purity, Crystallinity and Electromechanical Sensitivity of Lead-Free (Ba0.85Ca0.15)(Zr0.10Ti0.90)O3 Synthesized Using an EDTA/glycerol Modified Pechini Method"

_materials, 2025, doi:10.3390/ma18051180_

Round 1

Reviewer 1 Report

Comments and Suggestions for Authors

In this work, the authors prepared (Ba0.85Ca0.15)(Zr0.10Ti0.90)O3 crystal by an EDTA/glycerol modified Pechini method. The complete chemical transformation is achieved 300 °C lower than conventional synthesis methods. The prepared perovskite crystal shows promising electromechanical properties. However, some important issues should be carefully solved before acceptance.

1.     Full names should be given to all abbreviations when they first appear, such as EDTA, d33, d31, kp, 𝜀33𝑇, and Qm.

2.     To study the elemental distrobution, EDX mapping should be provided.

3.     Obvious shift of XRD peaks is observed in Figure 3c, which could be attributed to the appearance of structural strain. Please refer to the paper with DOI of 10.1063/5.0083059.

4.     This version lacks necessary electronic characterizations for the prepared sample, for example the detailed oxygen species can be extracted from the fitting of O 1s XPS. The authors can refer to the paper with DOI of 10.1002/cey2.465.

Reviewer 2 Report

Comments and Suggestions for Authors

The authors have reposted on a single phase material, (Ba0.85Ca0.15)(Zr0.10Ti0.90)O3, with a tetragonal structure, synthesized using a modified Pechini method and as a result they have achieved homogeneous grain size distribution. The manuscript is well prepared. However, there are a few suggestions to further improve this manuscript before publication.

The overall English would benefit form a proofreading.

Could the authors include elemental analysis of the materials synthesized, such as EDX elemental mapping?

Could the authors report on the porous size of the sintered ceramics?

Comments on the Quality of English Language

The overall English would benefit form a proofreading.

Reviewer 3 Report

Comments and Suggestions for Authors

1. It would be good to show photos of actual samples in Fig. 1.

2. There is a lack of analytical analysis to support the findings from experimental observations. 3. More in-depth comparison between the original and proposed modified Pechini method is needed. 4. More details are needed about how Table 3 is obtained. 5. What is the size of the sample? There is in general a lack of sufficient sample size to justify any variation.

Round 2

Reviewer 3 Report

Comments and Suggestions for Authors

Comments are addressed properly.

Author Response

Comment 1: Comments are addressed properly.

Dear Reviewer,

Thank you for your positive feedback and for taking the time to review our responses and manuscript. We appreciate your contribution to improving the quality of our work.

Best regards,

Sebastiano Garroni